# The Design of a Vision-Assisted Dynamic Antenna Positioning Radio Frequency Identification-Based Inventory Robot Utilizing a 3-Degree-of-Freedom Manipulator

**DOI:** 10.3390/s25082418

**Published:** 2025-04-11

**Authors:** Abdussalam A. Alajami, Rafael Pous

**Affiliations:** Department of Information and Communication Technologies Engineering, Pompeu Fabra University, 08002 Barcelona, Spain; rafael.pous@upf.edu

**Keywords:** RFID, inventory robot, dynamic inventory management, robotic hand, autonomous mobile robots, antenna manipulator, warehouse automation

## Abstract

In contemporary warehouse logistics, the demand for efficient and precise inventory management is increasingly critical, yet traditional Radio Frequency Identification (RFID)-based systems often falter due to static antenna configurations that limit tag detection efficacy in complex environments with diverse object arrangements. Addressing this challenge, we introduce an advanced RFID-based inventory robot that integrates a 3-degree-of-freedom (3DOF) manipulator with vision-assisted dynamic antenna positioning to optimize tag detection performance. This autonomous system leverages a pretrained You Only Look Once (YOLO) model to detect objects in real time, employing forward and inverse kinematics to dynamically orient the RFID antenna toward identified items. The manipulator subsequently executes a tailored circular scanning motion, ensuring comprehensive coverage of each object’s surface and maximizing RFID tag readability. To evaluate the system’s efficacy, we conducted a comparative analysis of three scanning strategies: (1) a conventional fixed antenna approach, (2) a predefined path strategy with preprogrammed manipulator movements, and (3) our proposed vision-assisted dynamic positioning method. Experimental results, derived from controlled laboratory tests and Gazebo-based simulations, unequivocally demonstrate the superiority of the dynamic positioning approach. This method achieved detection rates of up to 98.0% across varied shelf heights and spatial distributions, significantly outperforming the fixed antenna (21.6%) and predefined path (88.5%) strategies, particularly in multitiered and cluttered settings. Furthermore, the approach balances energy efficiency, consuming 22.1 Wh per mission—marginally higher than the fixed antenna (18.2 Wh) but 9.8% less than predefined paths (24.5 Wh). By overcoming the limitations of static and preprogrammed systems, our robot offers a scalable, adaptable solution poised to elevate warehouse automation in the era of Industry 4.0.

## 1. Introduction

Efficient inventory management constitutes a cornerstone of modern supply chain logistics, where the imperative for rapid, accurate, and autonomous product tracking has escalated alongside the demands of globalized commerce [1,2]. The evolution of robotic technology has been pivotal in addressing these challenges [3], transitioning from rudimentary automated guided vehicles (AGVs) reliant on fixed tracks and magnetic tapes to sophisticated Autonomous Mobile Robots (AMRs) empowered by advanced localization, mapping, and artificial intelligence (AI) systems. Early robotic solutions, constrained by static navigation paths and minimal environmental adaptability, have given way to a new generation of intelligent platforms capable of dynamic decision making, real-time spatial reasoning, and collaborative operation within complex warehouse ecosystems [4].

A critical breakthrough in this progression emerged with the integration of Simultaneous Localization and Mapping (SLAM) algorithms and high-precision sensors such as LiDAR, inertial measurement units (IMUs), and depth cameras. These technologies enabled robots to construct real-time maps of unstructured environments while concurrently estimating their position within them [5,6]. For inventory robots, SLAM eliminated dependence on preconfigured infrastructure, allowing navigation through dynamic aisles cluttered with pallets, personnel, and transient obstacles. Enhanced by probabilistic filtering techniques (e.g., Kalman and particle filters), modern SLAM frameworks achieve centimeter-level localization accuracy [7], ensuring reliable traversal and repeatable scanning operations even in densely populated warehouses.

The advent of AI and computer vision has further revolutionized robotic inventory systems, infusing them with unprecedented cognitive depth [8]. Deep learning architectures, such as convolutional neural networks (CNNs) and transformer models, empower robots to interpret visual data with human-like acuity. For instance, YOLO (You Only Look Once) object detection algorithms [9] enable real-time identification of items, enabling the possibility of RFID tag locations, while semantic segmentation models distinguish between product categories and storage configurations. These capabilities synergize with SLAM to optimize path planning, prioritize scan regions, and adapt to fluctuating inventory layouts [10].

Radio Frequency Identification (RFID)-enabled robots epitomize the convergence of these advancements, offering automated solutions to transcend the inefficiencies of manual inventory processes [11]. However, persistent challenges in tag detection accuracy, environmental adaptability, and operational endurance underscore the need for innovative solutions that further leverage these technological strides. While contemporary robots harness SLAM for robust navigation and AI for basic rfid-based inventory task execution, the next frontier lies in unifying these technologies into cohesive, context-aware platforms capable of dynamic sensor reconfiguration, adaptability to different warehouse configurations, and increasing the efficiency and performance of these machines.

This paper is structured as follows. Section 2 reviews existing state-of-the-art RFID-based inventory management approaches, including ground-based AMRs, aerial UAVs, hybrid systems, and advanced sensor-equipped AMRs, discussing their limitations. Section 3 details the hardware and software architecture of the proposed RFID-HAND robot, focusing on its manipulator, sensor suite, mobility system, and ROS2 framework. Section 5 presents comparative experiments that evaluate the performance of different scanning strategies, demonstrating the superiority of dynamic positioning. Section 6 compares the RFID-HAND robot with a state-of-the-art robot in a virtual warehouse, highlighting the advantages of dynamic antenna positioning. Section 7 outlines potential research directions, including algorithmic optimizations and expanded capabilities. Section 8 summarizes the key findings, emphasizing the potential of vision-assisted dynamic antenna positioning for enhanced RFID-based inventory management.

## 2. Related Work

Recent advancements in autonomous warehouse robotics for inventory management have converged around two critical axes: improving the mobility/navigation of warehouse inventory robots and the fusion of location/proximity sensors with RFID to increase the accuracy of inventory perception. Traditional systems, while effective in structured environments, struggle to reconcile the competing demands of dynamic obstacle avoidance, real-time inventory tracking, and energy efficiency. Five principal paradigms dominate the literature, each offering unique tradeoffs:

Ground-based fixed RFID antennas on AMRs equipped with static RFID antennas have been widely deployed and investigated by researchers for warehouses and retail environments [12,13,14,15]. These systems leverage ground mobility to navigate aisles, yet their fixed antenna configurations inherently limit spatial coverage, particularly in cluttered or densely stacked environments. Occluded tags and vertical blind spots necessitate labor-intensive manual verification, undermining the autonomy these systems aim to achieve.

Aerial inventory unmanned aerial vehicles (UAVs) address spatial limitations by deploying RFID sensors from elevated vantage points, as shown in various research [16,17,18]. While aerial platforms enhance accessibility to high-altitude storage areas and can navigate in hard-to-access zones for AMRs, they face critical constraints, including limited flight endurance (usually less than 30 min), navigational complexity for indoor environments, and reduced tag detection accuracy due to variable antenna–tag distances.

Hybrid ground–aerial systems: Recent research to harmonize aerial and terrestrial mobility has yielded hybrid robots capable of multimodal ground and areal operations in [19,20]. For example, Ref. [21] introduced a hybrid UAV-wheeled platform with custom brakes on board for 3D warehouse inventory tasks. Despite improved versatility, such systems grapple with energy inefficiency, mode-switching latency, and the need for intricate control architectures to manage heterogeneous locomotion dynamics.

Advanced sensor-equipped AMRs have been investigated by scientists developing cutting-edge platforms like the Dextory robot [22] that utilizes extendable sensor towers (e.g., 12 m masts) for vertical movement of RFID antennas and coverage, although at the cost of mechanical complexity and persistent blind spots to occlusion. In contrast, the research in [23] adopts a minimalist design, leveraging rotating antennas and LiDAR-guided navigation to achieve high-resolution 3D localization without bulky infrastructure. Its integration of visual odometry and depth cameras enables robust navigation in shop-like scenarios with metal racks and wooden shelves, although its static rotational patterns lack the real-time adaptability required for irregular tag distributions, especially tags placed at high shelves. AI-enhanced navigation systems recently have been investigated and adapted by scientists to increase the performance of these robots. The work in [24] exemplifies the shift toward AI-driven adaptability. By replacing conventional Q-learning with Deep Q-Networks (DQNs), their framework achieved a 40% improvement in learning efficiency, enabling robotic agents to dynamically optimize paths in response to fluctuating inventory layouts. While their YOLO-based object detection (95% accuracy) supplants legacy QR code scanning, the continued dependence on visual markers, a technology increasingly eclipsed by RFID in modern warehouses, limits scalability in retrofitted environments.

This paper introduces the RFID-HAND robot, a vision-assisted autonomous platform that features a 3-degree-of-freedom (3DOF) robotic manipulator for dynamic antenna positioning. Unlike prior systems reliant on static or preprogrammed configurations, our design uniquely synergizes real-time visual perception, adaptive manipulator kinematics, and RFID signal optimization to address the limitations of existing solutions.

The proposed RFID-HAND robot transcends these paradigms by unifying SLAM navigation, AI-vision-guided antenna manipulation and scanning, and adaptive RFID sensing. Unlike the robot developed in [24], which employs decision trees for post-detection analysis, our system eliminates intermediary processing through closed-loop control: YOLO detections directly inform inverse kinematic calculations, enabling precision antenna positioning without QR code dependencies. This integration resolves the perceptual latency inherent in hybrid architectures while surpassing the spatial constraints of rotating antenna systems. Furthermore, our ROS2-based navigation stack—augmented by SLAM and dynamic costmaps—parallels the policy gradient advantages of DQNs, achieving collision-free traversal in cluttered environments without sacrificing RFID scanning efficacy.

## 3. Robot Design

### 3.1. RFID-HAND Robot Hardware Description

The hardware description of the RFID-HAND robot can be categorized and explained into three blocks, as shown in Figure 1 and described in detail below:Robot arms block: The robot is designed with a robotic hand offering 3 DOF, which is powered by three cost-effective servo motors equipped with feedback capabilities for precise and accurate movement. An RFID antenna is mounted on the end effector, allowing for efficient spatial manipulation. The joints of the robotic hand are interconnected with an elastic rubber band, capable of supporting weights up to 30 kg, which not only reduces pressure on the servos but also enhances the stall torque limit, thus optimizing the performance of each joint.Sensors block: The sensor array of the robot is meticulously arranged, featuring a 2D LiDAR sensor mounted on top of the chassis for critical proximity detection and 2D environmental mapping. Additionally, an RGB-D camera is integrated to capture both color images and depth information, enabling the robot to accurately perceive the shape, size, and distance of objects in its vicinity. The computational backbone of the robot is a cost-effective and energy-efficient single-board computer (SBC), specifically the n100, which processes sensor data, executes control algorithms, and supports advanced functionalities such as neural network-based object recognition, navigation, and localization.Mobility block: For mobility, the robot is equipped with two 24 V high-torque motors, each fitted with hall sensors and a motion controller, ensuring precise control over the robot’s movement, velocity, and acceleration. The power system is designed for sustained operation, either via a rechargeable battery pack or an external power source, ensuring the robot’s continuous functionality.

### 3.2. RFID-HAND Robot Software Description

The software architecture deployed on the robotic platform is centered around the Robot Operating System 2 (ROS2) framework [25], which provides a robust and scalable foundation for complex robotic operations. The integration of the differential drive motor controller with ROS2’s control software [26] facilitates precise communication, enabling accurate odometry and controlled locomotion.

The robot’s navigation system utilizes the ROS2 Nav2 stack [27], a multilayered software that enables the creation of both local and global costmaps. These costmaps are derived from sensor data gathered by the robot’s comprehensive perception layer, allowing for sophisticated interaction with the environment. The perception layer is a critical component, comprising the data obtained from the array of sensors. This sensor suite includes 360-degree LiDAR data for comprehensive proximity data, an RGB-D depth camera point cloud data for enhanced environmental awareness through depth perception, and localization sensor data, such as from the hall sensors integrated into the motors and an inertial measurement unit (IMU), which contribute to the robot’s spatial orientation and map generation. Collectively, these sensory inputs provide the necessary data to construct and refine maps, which are essential for effective navigation and obstacle avoidance. The navigation and path-planning module leverages data from the perception layer to compute optimal trajectories toward predefined goals while dynamically avoiding obstacles. This capability is achieved through path-planning algorithms that process real-time sensor data to assess the environment, analyze current conditions, and determine the most efficient and safe paths for the robot’s movement. In addition to navigation, the software system includes a custom-designed controller for the robotic arms mounted on the platform. This controller acts as an intermediary, integrating inverse kinematics algorithms from ROS2 MoveIt [28] with essential data on joint angles, velocities, and feedback for precise control of the robot’s arms. Through its interface with ROS2 MoveIt, the custom controller translates high-level motion plans into executable commands, enabling the robot’s arms to operate with accuracy and responsiveness. Moreover, the robot’s software system incorporates an algorithm that utilizes detection information from a pretrained YOLO model [9] to identify potential warehouse items within its environment. Upon detecting an item, the algorithm computes the optimal kinematic movements for the robot’s 3DOF arm, which is equipped with an RFID antenna at its end effector. The algorithm precisely orients the arm toward the detected item, ensuring accurate targeting. Once aligned, the arm performs a controlled circular motion around the item’s perimeter, effectively covering its entire surface area to maximize RFID scanning efficiency and ensure comprehensive data capture of the item.

## 4. Technical Overview

The robot employs a pretrained YOLO9 model to detect boxes and similar warehouse products that are attached to RFID tags. YOLO9 provides bounding boxes and pixel positions of detected objects in the camera’s image plane. To interact with these objects, the robot translates the 2D pixel positions into 3D world coordinates utilizing the information of the depth point supplied by the camera, enabling the robotic arm to orient its end effector accurately. The depth value $g_{z}$ is obtained from the RGB-D depth camera corresponding to the center pixel $\left(x_{c},y_{c}\right)$. The camera’s intrinsic matrix *K* is defined using Equation (Equation 1):(1)$$K=\left[\begin{array}{ccc} f_{x} & 0 & c_{x}\\ 0 & f_{y} & c_{y}\\ 0 & 0 & 1 \end{array}\right]$$
where $f_{x}$ and $f_{y}$ are the focal lengths, and $\left(c_{x},c_{y}\right)$ is the optical center. For a detected object’s bounding box with the top-left corner $\left(x_{\min},y_{\min}\right)$ and the bottom-right corner $\left(x_{\max},y_{\max}\right)$, the center of the bounding box in pixel coordinates and the 3D world coordinates $\left(g_{x},g_{y},g_{z}\right)$ of the detected object can be derived in Equations (Equation 2) and (Equation 3):(2)$$x_{c}=\frac{x_{\min}+x_{\max}}{2},y_{c}=\frac{y_{\min}+y_{\max}}{2}$$(3)$$g_{x}=\frac{\left(x_{c}-c_{x}\right)\cdot g_{z}}{f_{x}},g_{y}=\frac{\left(y_{c}-c_{y}\right)\cdot g_{z}}{f_{y}}$$

The accuracy of object detection using YOLO can be affected by varying lighting conditions and occlusions commonly encountered in warehouse environments. Such factors may lead to missed detections or inaccurate bounding boxes, subsequently impacting antenna positioning accuracy. To mitigate these issues, we employed data augmentation techniques during YOLO training that specifically included varied lighting conditions and partial occlusions to enhance robustness against real-world environmental variations. A position of the center of a detected object is therefore extracted with coordinates $g_{x},g_{y},g_{z}$. Using the trigonometry equations in Equations (Equation 4)–(Equation 8) and as illustrated in Figure 2, we compute the position $h_{x},h_{y},h_{z}$ along the line $\vec{r}$. *h* is the desired point position to move the end effector (RFID antenna) to a point at the center of the detected object.(4)$$n=\sqrt{l_{2}^{2}-h_{z}^{2}}$$(5)$$\tan\theta =\left(\frac{m}{o}\right)=\left(\frac{h_{z}}{l_{1}+n}\right)$$(6)$$h_{z}=\left(\frac{m(l_{1}+\sqrt{l_{2}^{2}-h_{z}^{2}})}{o}\right)$$$$=h_{z}=\sqrt{\left(\frac{m^{2}\cdot l_{2}^{2}+m\cdot l_{1}}{o^{2}+m^{2}}\right)}$$(7)$$\tan\phi =\left(\frac{h_{y}}{t}\right)=\left(\frac{p}{r}\right)$$(8)$$h_{x}=l_{1}+n$$
where $l_{1}$ is the distance between the first joint in M1 (called the arm base) and the second joint at M2 (called the shoulder joint). $l_{2}$ is the distance between the second joint in M2 and the third joint in M3 (called the elbow joint). $l_{3}$ is the distance between the third joint in M3 and the end effector (the antenna).

$\theta_{1}$, $\theta_{2}$, and $\theta_{3}$ are the angles of the y-axis plane joints M1, M2, and M3, respectively, as shown in Figure 2.

$\phi_{1}$, $\phi_{2}$, and $\phi_{3}$ are the angles of the joint of the x-axis planes of M1, M2, and M3, respectively, as shown in Figure 2. We assume that $\theta_{3}$, which is the angle of the elbow joint shown in Figure 2, is 0deg, as the robotic arm should be extended to the maximum for forming a pointing shape toward a target.

For further manipulation of the robotic arm, a forward and inverse kinematics model is used to move the frames as $M_{1}\to M_{2}\to M_{3}$, as shown in Figure 2. To represent the kinematic chain, we use the Denavit–Hartenberg (DH) [29] convention to define each joint’s transformation matrix. Using the DH parameters, the transformation matrices for each joint T1, T2, and T3 are respectively represented as Equations (Equation 9)–(Equation 11):(9)$$T1=\left[\begin{array}{cccc} \cos(\theta_{1}) & -\sin(\theta_{1}) & 0 & l_{1}\\ \sin(\theta_{1}) & \cos(\theta_{1}) & 0 & 0\\ 0 & 0 & 1 & 0\\ 0 & 0 & 0 & 1 \end{array}\right]$$(10)$$T2=\left[\begin{array}{cccc} \cos(\theta_{2}) & 0 & \sin(\theta_{2}) & l_{2}\cos\left(\theta_{2}\right)\\ 0 & 1 & 0 & 0\\ -\sin(\theta_{2}) & 0 & \cos(\theta_{2}) & l_{2}\sin\left(\theta_{2}\right)\\ 0 & 0 & 0 & 1 \end{array}\right]$$(11)$$T3=\left[\begin{array}{cccc} \cos(\theta_{3}) & -\sin(\theta_{3}) & 0 & l_{3}\cos\left(\theta_{3}\right)\\ \sin(\theta_{3}) & \cos(\theta_{3}) & 0 & l_{3}\sin\left(\theta_{3}\right)\\ 0 & 0 & 1 & 0\\ 0 & 0 & 0 & 1 \end{array}\right]$$

The overall transformation from the base to the end effector is given by Equation (Equation 12):(12)T=T1·T2·T3

Inverse kinematics involves determining the joint angles $\left(\theta_{1},\theta_{2},\theta_{3}\right)$ given the desired position (x,y,z) in the Cartesian coordinate system and can be extracted using the previous equations.

From the obtained transformation matrix *T*, the position of the end effector is defined as $R^{1\times 3}$.

The computed joint angles $\left(\theta_{1},\theta_{2},\theta_{3}\right)$ are fed into the robotic arm controller, which adjusts the arm’s position dynamically. To optimize RFID tag scanning and inventory management, the robot ensures that the RFID antenna at the end effector points toward and entirely covers the detected object. To move the end effector in a circular motion around the product, the robot must adjust the angles $\theta_{1}$, $\theta_{2}$, and $\theta_{3}$ dynamically. The circle’s radius must be appropriate to cover the product surface, with the center at $g_{x},g_{y},g_{z}$. From the given boundary box of the detected item, the distance between the maximum edge of the boundary box to the center would be considered as the radius *r*. The desired circular motion path that the end effector would need to follow is a set of consecutive points defined as $x^{\prime },y^{\prime },z^{\prime }$. The parametric equations for the circular path are shown in Equation (Equation 13):(13)$$x\left(t\right)=x^{\prime }+r\cos\left(t\right),y\left(t\right)=y^{\prime }+r\sin\left(t\right)$$
where *t* ranges from 0 to 2π.

## 5. Experiments

### 5.1. Short Aile Low Shelves Scanning

The objective of the experiment described here in Section 5.1 is to assess the comparative effectiveness of three RFID scanning approaches for inventory management. This study specifically evaluates the performance of the RFID-HAND robot using a static RFID antenna versus a predefined path strategy, where the manipulator moves the antenna across preset spatial points in addition to an intelligent detection and dynamic positioning method, where a pretrained YOLO model identifies probable products and utilizes forward and inverse kinematics to precisely position the manipulator’s end effector (the antenna) to perform a tailored circular motion around the object, ensuring comprehensive RFID tag scanning. The primary metric of comparison is the number of RFID tags detected while scanning shelves containing tagged boxes in a controlled environment. The experiment was carried out in a controlled laboratory with 15 m × 5 m layout dimensions. Low shelves with (0.2 m height) and high shelves (up to 1.5 m height) were placed along a straight line in the laboratory. A total of 500 RFID tags were distributed across shelf levels. The distribution of RFID tags were evenly placed inside boxes arranged systematically along shelves. The optimal RFID antenna reading range is (0–3 m), and the specific RFID reader power settings specified (30 dBm). The robot navigated a given path while placed 0.8 m away from the shelves to simulate a real warehouse aisle. The robot conducted two distinct experiments in the scenario illustrated in Figure 3.

#### 5.1.1. Fixed Antenna

In the experiment detailed here in Section 5.1.1, the robot was programmed to navigate in a straight line along the aisle, maintaining a constant speed. The antenna was fixed in one position oriented perpendicular to the shelves to scan for RFID tags, as shown in Figure 4. The robot made a single pass navigating along the aisle, recording the number of RFID tag reads.

#### 5.1.2. Articulated Antenna

In the experiment described here in Section 5.1.2, the robot was programmed to navigate the same straight line path along the aisle at the same speed as in experiment Section 5.1.1. As the robot moved, the robotic arm articulated in 3D space, moving the antenna in a sweeping motion to cover a broader spatial plane. The motion pattern of the arm was preprogrammed to ensure consistent coverage, as shown in Figure 5. The robot made a single pass along the aisle, recording the number of RFID tag reads.

Upon completion of both experiments, the RFID read data were analyzed to determine the number of unique RFID tags detected in each scenario. The results in Figure 6 demonstrate a significant difference in the number of RFID tags detected between the two scanning methods. The fixed antenna position resulted in a limited detection range, with a total of 117 tags read; however, the articulated robotic arm significantly increased the coverage area, resulting in 438 tags read from 500 total placed tags in the environment.

The articulated antenna setup showed a higher number of RFID reads, indicating improved spatial coverage and detection capability. This improvement is attributed to the dynamic motion of the antenna, which allowed better orientation and positioning relative to the tags.

#### 5.1.3. Dynamic Movement Articulated Antenna

In this experiment, the RFID-HAND robot utilized the vision-assisted dynamic antenna positioning method to evaluate its performance in reading RFID tags. The experimental setup was identical to that of the experiments described in Section 5.1.1 and Section 5.1.2.

The vision-assisted dynamic antenna positioning method integrated a pretrained YOLO model for object detection, which identified tagged boxes in real time. Using forward and inverse kinematics, the robot dynamically adjusted its 3DOF manipulator to align the RFID antenna with each detected box, as shown in Figure 7. The antenna then performed a tailored circular scanning motion around the object’s surface to maximize RFID tag detection coverage, as described in Section 4.

This experiment resulted in the detection of 479 unique tags out of the 500 placed tags, achieving a detection rate of 95.8%. The results are illustrated in Figure 8 alongside those from the experiments described in Section 5.1.1 and Section 5.1.2 for comparison.

The results demonstrate that incorporating vision-based dynamic positioning significantly enhances RFID tag detection performance compared to static or preprogrammed scanning methods. The ability to read 479 out of 500 tags demonstrates that this method effectively compensates for challenges such as occlusions, varied object heights, and complex spatial arrangements often encountered in warehouse environments.

### 5.2. Tall Ailes High Shelves Scanning

The objective of this experiment described in Section 5.2 is to compare the effectiveness and performance of the two different RFID scanning methods for inventory management as in the experiments described in Section 5.1.1, Section 5.1.2 and Section 5.1.3. The experiment was carried out in a controlled laboratory with 15 m × 5 m layout dimensions. Low shelves with (0.2 m height) and high shelves (up to 2.5 m height) were placed along a straight line in the laboratory. A total of 350 RFID tags were distributed across the shelf levels. The distribution of RFID tags were evenly placed inside and on the boxes arranged systematically along shelves. The optimal RFID antenna reading range is (0–3 m), and the specific RFID reader power settings specified (30 dBm).

The robot navigated a given path while placed 0.8 m away from the shelves to simulate a real warehouse aisle, as shown in Figure 9. The robot conducted two separate experiments.

#### 5.2.1. Fixed Antenna

As in experiment Section 5.1.1, the robot also was programmed to navigate in a straight line along the aisle, maintaining a constant speed. The antenna was fixed in one position oriented perpendicular to the shelves to scan for RFID tags. The robot made a single pass along the aisle, recording the number of RFID tag reads from both high and low shelves.

#### 5.2.2. Articulated Antenna

As in the experiment described in Section 5.2.1, the robot was programmed to navigate the same path and with the same speed along the aisle at the same speed as in the experiment in Section 5.2.1. As the robot moved, the robotic arm articulated in 3D space, moving the antenna in a sweeping motion to cover both high and low shelves as shown in Figure 10. The same preprogrammed motion pattern of the arm was used to ensure consistent coverage of all shelf levels. The robot made a single pass along the aisle, recording the number of RFID tag reads from both high and low shelves, as shown in Figure 11.

The fixed antenna position as in the experiment described in Section 5.2.1 resulted in a limited detection range, with a total of 67 tags read from high and low shelves. The programmed motion of the articulated arm increased the coverage area, resulting in a total of 314 RFID tags read from shelves from a total of 350 RFID tags placed, which shows a remarkable difference.

#### 5.2.3. Dynamic Movement Articulated Antenna

For the experiment described here in Section 5.2.3, the robot was programmed to navigate the same straight-line path along the aisle at the same speed as in previous experiments. As the robot moved, utilizing the information from the detections from the YOLO model, the algorithm dynamically guided the robotic arm, adjusting the antenna’s main radiation lobe toward the detected boxes with RFID tags in real time and then performing a controlled circular motion around the item’s perimeter, as shown in Figure 12, effectively covering its entire surface area, as explained in Section 3.2. The robot made a single pass along the aisle, recording the number of RFID tag reads from both high and low shelves.

The results demonstrated in Figure 13 show significant differences in the number of RFID tags detected between the three scanning methods, particularly for the high shelves.

The vision-assisted dynamic antenna positioning further increased the coverage area. The approach outperformed the others, resulting in 343 tags read from all shelves.

Table 1 summarizes the RFID tag detection rates and energy consumption among the scanning methods. The dynamic positioning approach achieved detection rates of 95.8% (479/500) and 98.0% (343/350) for low and high shelves, respectively, outperforming the fixed (23.4%, 19.1%) and predefined path (87.6%, 89.7%) methods. Notably, while the dynamic method consumed 22.1 Wh per mission and was marginally higher than the fixed antenna (18.2 Wh), it reduced energy use by 9.8% compared to the predefined paths method (24.5 Wh), demonstrating a balance between efficiency and performance.

## 6. Simulation Experiment: Dynamic vs. Fixed Antenna Performance

To quantitatively evaluate the advantages of dynamic antenna positioning, we conducted a comparative simulation of the proposed robot in this paper vs. a SOTA RFID-based inventory robot [30] in a Gazebo-based virtual warehouse environment.

The simulated workspace comprised three industrial pallet racks (2.4 m height × 5.2 m depth), each containing three horizontal tiers (0.8 m vertical spacing) replicating common warehouse storage configurations. A total of 600 passive UHF RFID tags were distributed pseudo-randomly across pallet surfaces using a uniform spatial distribution model, with tag orientations randomized to emulate real-world product placement scenarios. The tags were simulated using the RFID Gazebo plugin [31], which implements a probabilistic model specially designed for environments with large constellations of RFID tags, as described in [32]. Undetected tags were visually represented as gray cuboids (5 cm × 5 cm × 5 cm) to facilitate post-experiment analysis, as shown in Figure 14.

The experiment compared our RFID-HAND robot with the Robin50 platform, which is an RFID-based inventory robot employing a four static antenna configuration. To ensure equitable comparison, both systems operated with two active antennas: RFID-HAND utilized its dual-arm manipulators (one antenna per end effector), while Robin50’s left and right chassis-mounted antennas were activated, deactivating its front/rear arrays to match the antenna count. As illustrated in Figure 14, robots traversed a 78 m predefined path (0.8 m aisle clearance) at 0.2 m/s, adhering to ISO 3691-4 safety standards for warehouse AMRs.

Key performance metrics focused on tag detection rate versus temporal progression, with results shown in Figure 15. When the RFID-HAND robot detects an RFID tag, the tag changes its color to a red cube, as shown in Figure 16. The RFID-HAND achieved 97.1% detection accuracy (582/600 tags), demonstrating superior performance through its vision-guided adaptive scanning. In contrast, when the Robin50 robot detects an RFID tag, the tag changes its color to a yellow cube, as shown in Figure 16.

Robin50’s fixed antennas detected only 70.6% (424/600), which were primarily limited by the following:Vertical Coverage Constraints: Static antennas failed to maintain optimal read zones for upper-tier tags (1.6–2.4 m height), exhibiting 58.2% detection rate above 1.2 m versus 96.4% for RFID-HAND.Occlusion Sensitivity: Fixed beam patterns could not circumvent pallet obstructions, whereas the manipulator actively reoriented antennas to exploit RF propagation paths.Angular Coverage Limitations: The circular scanning motion increased antenna–tag polarization alignment opportunities, raising read likelihood by around 26.5% compared to fixed orientations.

The results demonstrated in Figure 13, show significant differences in the number of RFID tags detected between the three scanning methods, particularly for high shelves. These results empirically validate that dynamic antenna positioning substantially outperforms static configurations in cluttered, multitiered environments. The 26.5% performance gap highlights the critical importance of spatial adaptability in RFID-based inventory systems, particularly when scanning irregularly arranged stock-keeping units. This experiment’s methodology and dataset have been open sourced to facilitate reproducibility and benchmarking efforts in autonomous inventory robotics.

## 7. Future Work

Future research will focus on further refinement of the robot’s object detection and antenna positioning algorithms to enhance accuracy and reduce processing time. Additionally, we aim to explore the integration of machine learning techniques to adapt the scanning patterns based on real-time feedback, potentially improving performance in highly cluttered environments. From the experiments, we observed that an optimal robot velocity of approximately 0.2 m/s ensures both accurate RFID tag readings and safe interactions within dynamic environments. At this speed, the processing time required by the YOLO model for object detection and subsequent real-time kinematic calculations is sufficient to maintain robust performance without noticeable delays. Nonetheless, we recognize that computational latency could pose limitations if higher operational speeds were desired. Therefore, future research will explore algorithmic optimization techniques such as hardware acceleration (e.g., GPU integration) or lightweight neural network architectures to further reduce computational delays and potentially enable higher speed operations without compromising accuracy or safety. Another area of interest is the expansion of the robot’s capabilities to handle a broader range of object shapes and materials, as well as the development of multirobot systems for cooperative inventory tasks. Ultimately, although our current experiments demonstrated significant advantages of dynamic antenna positioning under simplified controlled scenarios, future research will involve rigorous testing within real warehouse environments and more complex controlled settings featuring densely packed shelves, dynamic obstacles such as moving personnel or vehicles, partial occlusions, varied lighting conditions, and realistic operational constraints.

## 8. Conclusions

In this study, we presented the design and implementation of a vision-assisted dynamic antenna positioning RFID-based inventory robot that utilizes a 3DOF manipulator. The robot integrates a 3DOF arm with an RFID antenna in its end effector, making it capable of detecting and scanning objects within a warehouse environment. Our approach leverages both forward and inverse kinematics to precisely position the antenna, allowing for a comprehensive scan of RFID tags through circular motion tailored to the object’s size. The comparative analysis demonstrated that our proposed method outperforms traditional fixed antenna configurations and predefined movement strategies in terms of tag detection efficiency and coverage. Specifically, the dynamic positioning and targeted scanning enabled by YOLO-based object detection significantly increased the number of RFID tags read across various object sizes and heights, marking a substantial advancement in inventory management robotics. Furthermore, the dynamic method achieved superior energy efficiency, consuming 22.1 Wh per mission compared to 24.5 Wh for predefined paths, thus optimizing detection performance while maintaining sustainable operational costs in inventory tasks. The simulation results demonstrate that dynamic antenna positioning significantly enhances RFID detection efficacy, with the proposed system achieving 97.1% tag detection versus 74.0% for static configurations. This 26.5% improvement highlights the value of adaptive spatial reconfiguration in addressing vertical coverage gaps, occlusions, and polarization mismatches in multitiered environments. Vision-guided manipulation enables precise alignment, particularly for elevated (>1.2 m) or obstructed tags. While these findings suggest substantial potential for real-world applications, further validation through physical deployments in operational warehouses is necessary to confirm practical scalability and robustness. Although our experiments demonstrated significant improvements under controlled laboratory conditions, future work will involve rigorous testing in more complex warehouse environments featuring densely packed shelves, dynamic obstacles such as moving personnel or vehicles, partial occlusions, varied lighting conditions, and realistic operational constraints to thoroughly validate our system’s robustness.

## Figures and Tables

**Figure 1 sensors-25-02418-f001:**
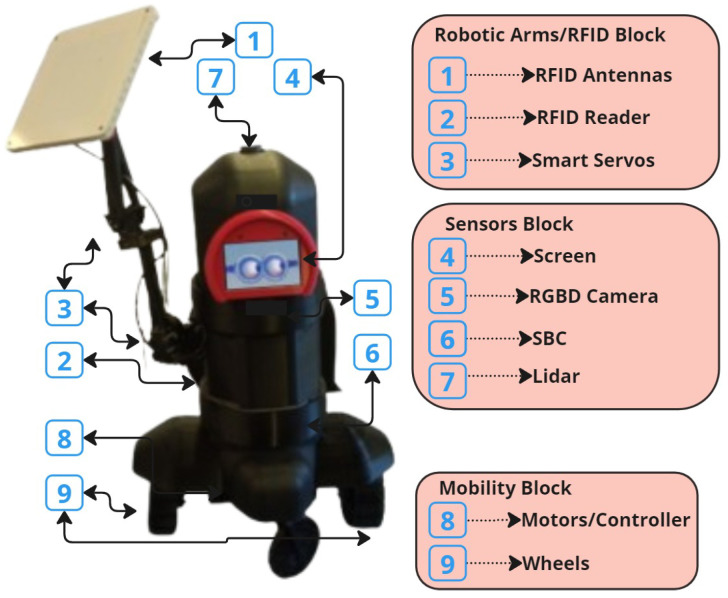
RFID-HAND hardware description.

**Figure 2 sensors-25-02418-f002:**
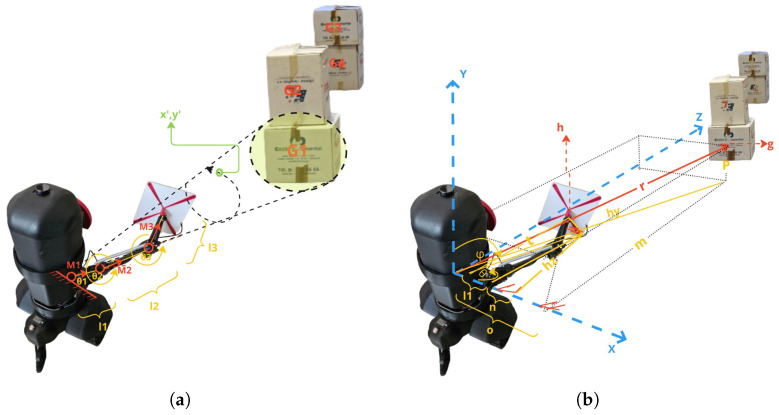
(**a**) Joint configurations and circular scanning motion around detected product. (**b**) Trigonometric parameters used for end effector positioning.

**Figure 3 sensors-25-02418-f003:**
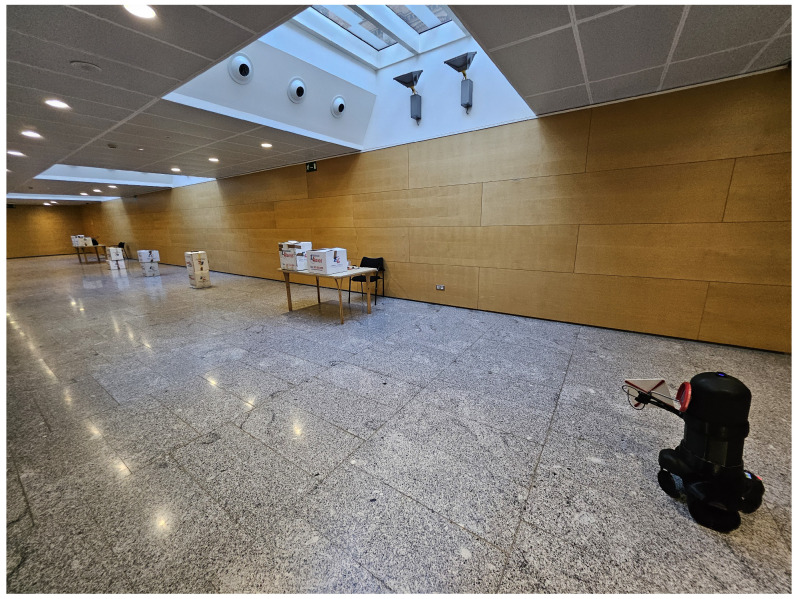
Controlled warehouse scenario illustrating shelf arrangement with clearly defined shelf heights (0.2 m low shelves; 1.5 m high shelves), aisle dimensions (15 m length × 5 m width), and systematic RFID tag distribution.

**Figure 4 sensors-25-02418-f004:**
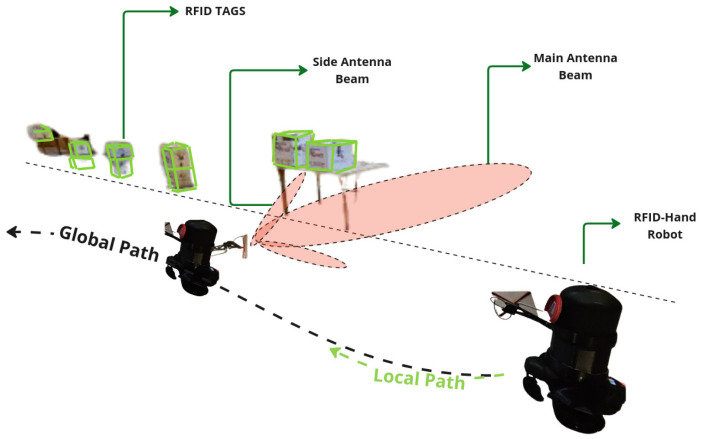
A descriptive illustration of the RFID-HAND robot performing inventory with a fixed antenna in experiment Section 5.1.1.

**Figure 5 sensors-25-02418-f005:**
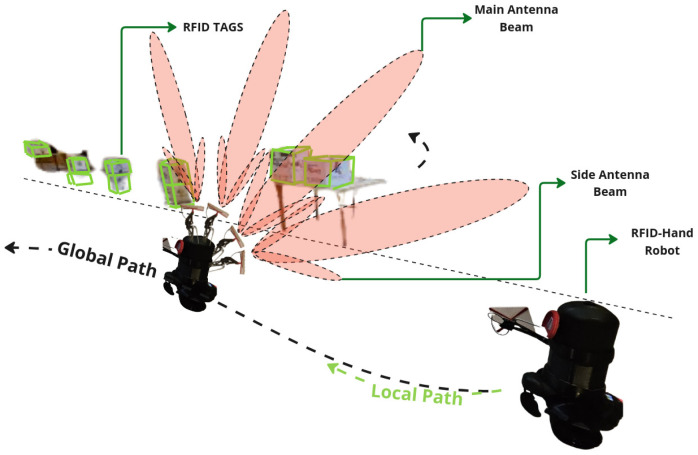
A descriptive illustration of the RFID-HAND robot performing inventory with an articulated antenna in experiment Section 5.1.2.

**Figure 6 sensors-25-02418-f006:**
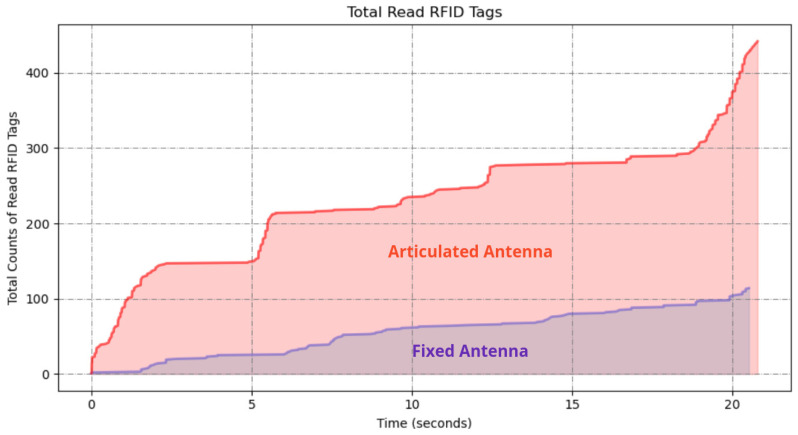
A graph of RFID tag readings from the robot with fixed antenna and articulated antenna experiments.

**Figure 7 sensors-25-02418-f007:**
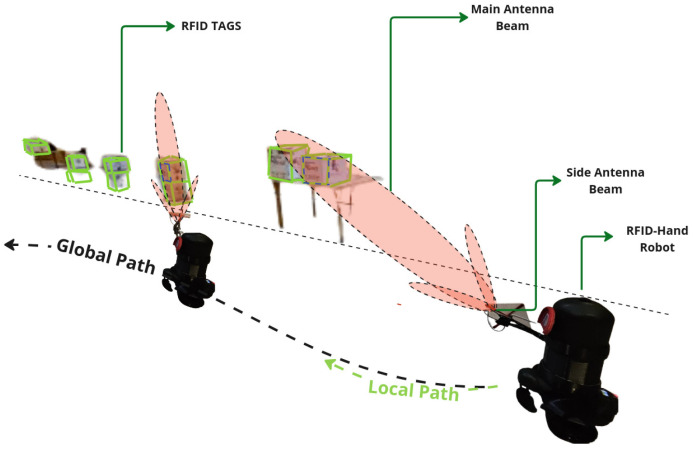
A descriptive illustration of the RFID-HAND robot performing inventory with an dynamic moving articulated antenna in experiment in Section 5.1.3.

**Figure 8 sensors-25-02418-f008:**
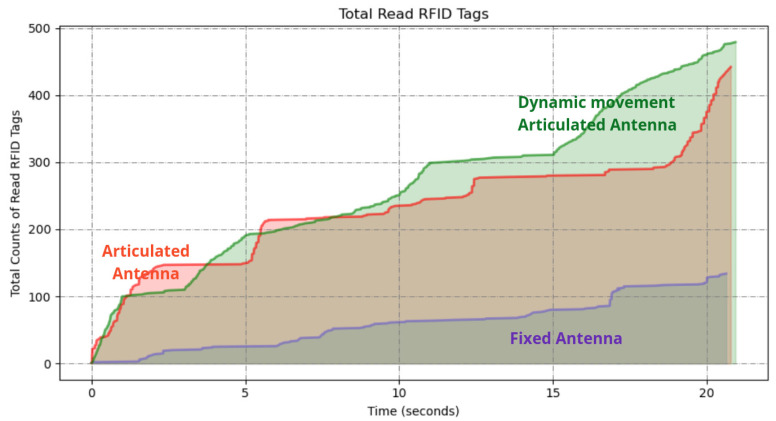
A graph of the resulting RFID tag readings from the robot with fixed antenna in experiment in Section 5.1.1, articulated antenna in experiment in Section 5.1.2, and articulated antenna in experiment in Section 5.1.3.

**Figure 9 sensors-25-02418-f009:**
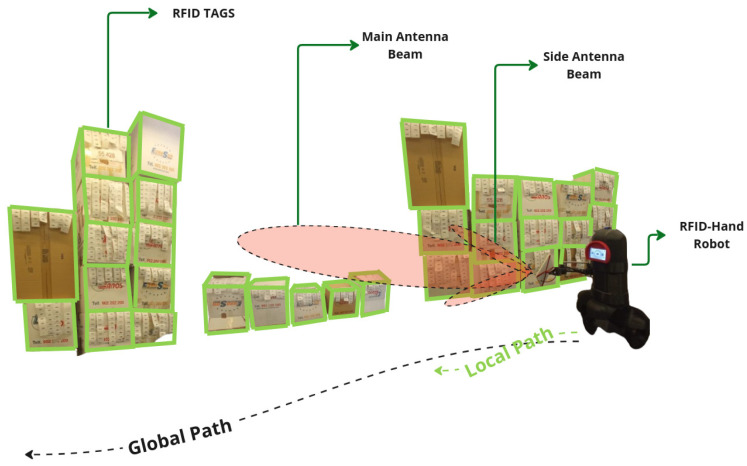
A descriptive illustration of the RFID-HAND robot performing inventory with a fixed antenna in experiment in Section 5.2.1.

**Figure 10 sensors-25-02418-f010:**
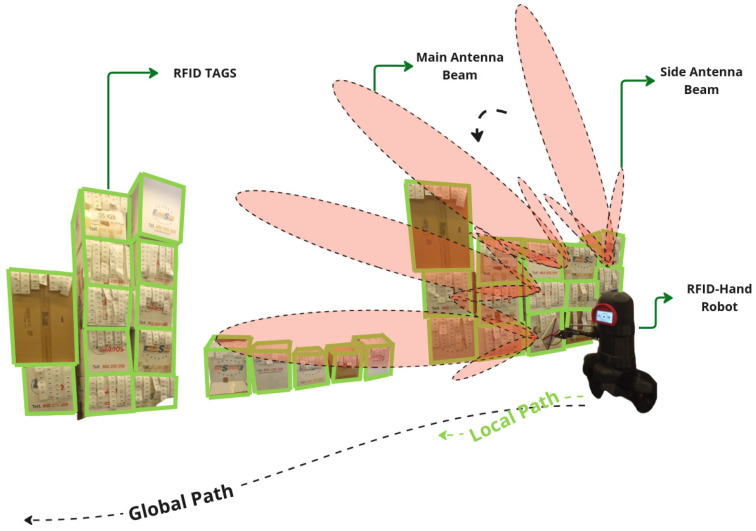
A descriptive illustration of the RFID-HAND robot performing inventory with an articulated antenna in experiment in Section 5.2.2.

**Figure 11 sensors-25-02418-f011:**
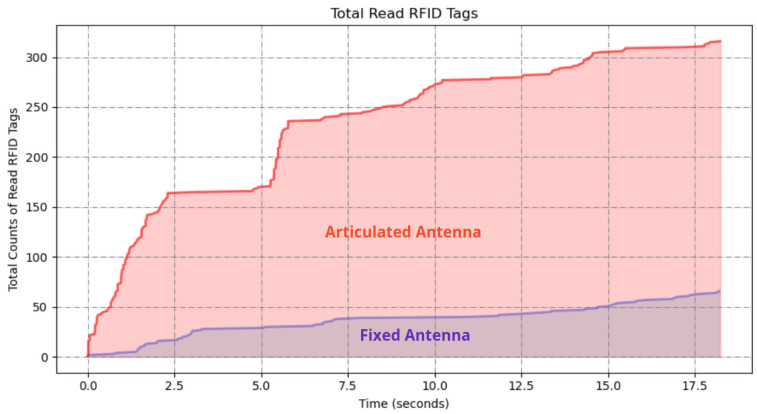
A graph of the resulting RFID tag readings from the robot with fixed antenna in experiment in Section 5.2.1 and articulated antenna in experiment in Section 5.2.2.

**Figure 12 sensors-25-02418-f012:**
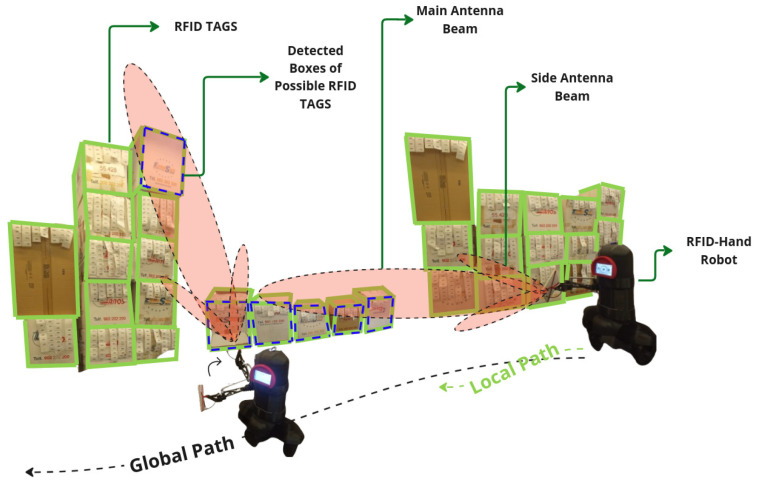
A descriptive illustration of the RFID-HAND robot performing inventory with an dynamic moving articulated antenna in experiment in Section 5.2.3.

**Figure 13 sensors-25-02418-f013:**
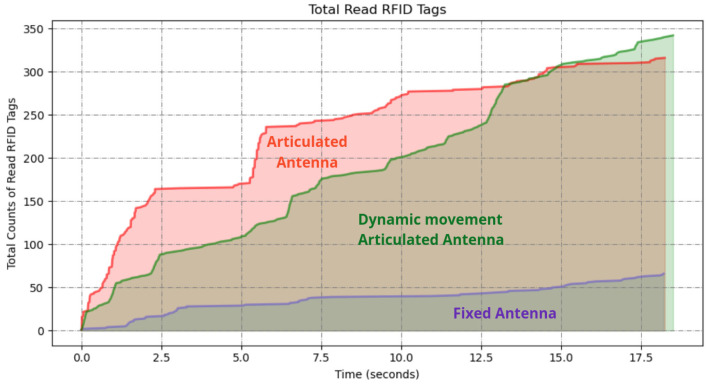
A graph of the resulted RFID tag readings from the robot with fixed antenna in experiment in Section 5.2.1, articulated antenna in experiment in Section 5.2.2, and dynamic movement antenna in experiment in Section 5.2.3.

**Figure 14 sensors-25-02418-f014:**
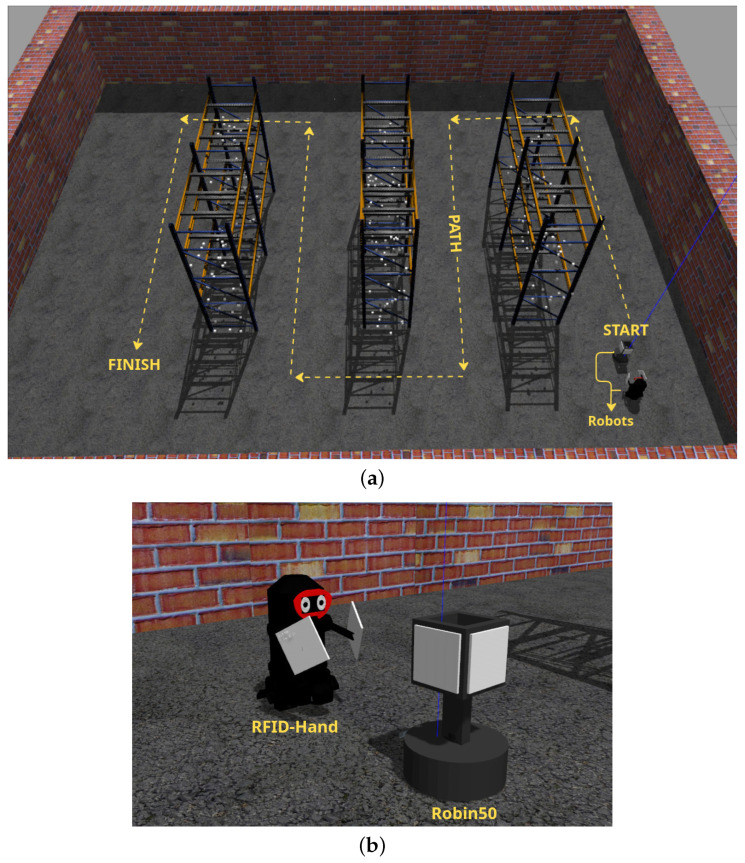
(**a**) Simulated robots in gazebo simulator. (**b**) Simulated warehouse in Gazebo simulator.

**Figure 15 sensors-25-02418-f015:**
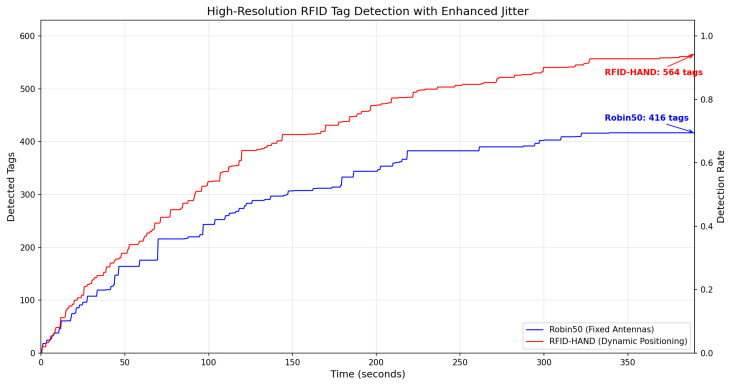
The plot of the detected RFID tags vs. time obtained by both robots from the simulation experiment.

**Figure 16 sensors-25-02418-f016:**
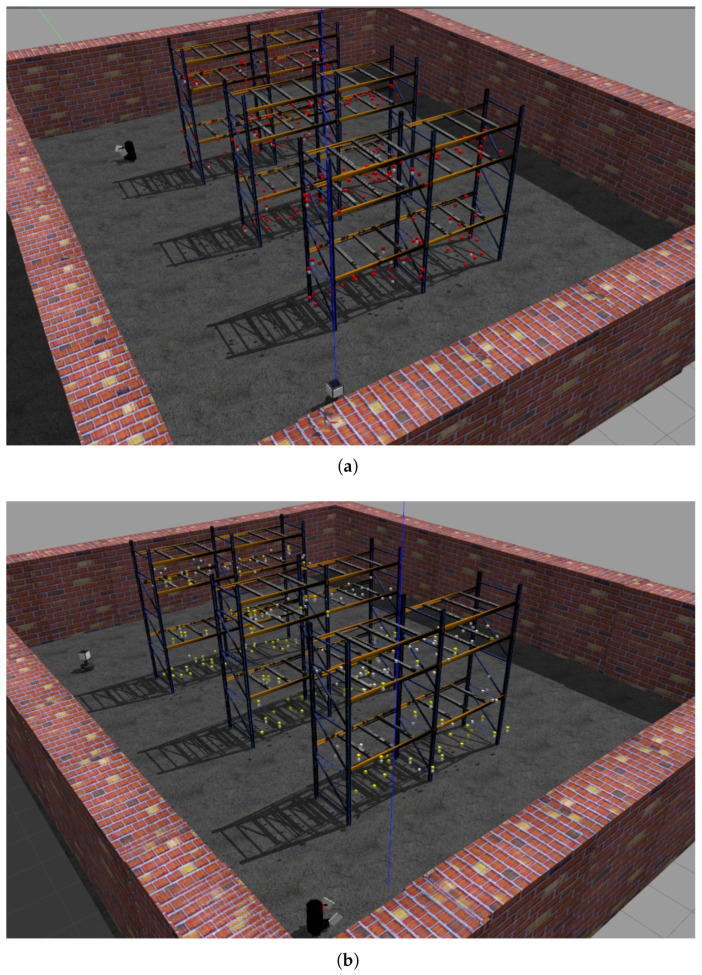
(**a**) Illustration of the RFID-HAND robot with the detected RFID tags (in red boxes) after successfully navigating the given path. (**b**) Illustration of the Robin50 with the detected RFID tags (in yellow boxes) after successfully navigating the given path.

**Table 1 sensors-25-02418-t001:** Comparative performance and energy efficiency of scanning methods.

Scanning Method	Tags Read		Total Read (%)	Energy Used (Wh)
	Low Shelves	High Shelves		
Fixed Antenna	117/500 (23.4%)	67/350 (19.1%)	21.6	18.2
Predefined Path	438/500 (87.6%)	314/350 (89.7%)	88.5	24.5
Dynamic Positioning	479/500 (95.8%)	343/350 (98.0%)	96.7	22.1

## Data Availability

Data are contained within the article.

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
