# Peer review of "The Design of a Vision-Assisted Dynamic Antenna Positioning Radio Frequency Identification-Based Inventory Robot Utilizing a 3-Degree-of-Freedom Manipulator"

_sensors, 2025, doi:10.3390/s25082418_

Round 1
Reviewer 1 Report
Comments and Suggestions for Authors
-
1)In the " Abstract.", for the first occurrence of the abbreviation "RFID", it needs to list the full name.
2)Key words missing in the paper, please add.
3)In 109 line RGBD depth camera format as well as 139 lines RGB-D, should remain the same in the full text.
4)This manuscript describes the design and evaluation of an RFID based inventory robot, and uses visual and 3 degree of freedom (DOF) manipulators for dynamic antenna positioning. However, the simulation experiment data is less, and it is suggested to add the experiment of verification algorithm.
5)The signs n, θ, Ï•,l1 and l2 in formula (4) to (8) are not obvious on the figure. Please explain their specific meaning in the text to avoid ambiguity.
6)In "4. Experiments", the experimental environment shown in Figure 3 is relatively vague, which can be combined with the setting of the experimental environment (such as warehouse layout, shelf height, distribution of RFID tags, etc.) and the specific parameters of the robot and RFID equipment (such as the reading range and power of RFID antenna, etc.).
The English could be improved to more clearly express the research.
The subtitle of Figure 2 can be succinct.
Author Response
Please see attached .doc file.
Thank you
Response to Reviewer 1 Comments
|
||||||||||||||||||||
1. Summary |
|
|
||||||||||||||||||
We sincerely appreciate Reviewer 1's valuable feedback and constructive suggestions, which have significantly improved the clarity and quality of our manuscript. Below, we provide detailed responses to each comment, clearly indicating the changes made.
|
||||||||||||||||||||
2. Questions for General Evaluation
3. Point-by-point response to Comments and Suggestions for Authors |
||||||||||||||||||||
Comments 1: In the " Abstract.", for the first occurrence of the abbreviation "RFID", it needs to list the full name. |
||||||||||||||||||||
Response 1: Agree. We have revised the abstract to explicitly state "Radio Frequency Identification (RFID)" at first occurrence. |
||||||||||||||||||||
Comments 2: Key words missing in the paper, please add. |
||||||||||||||||||||
Response 2: Agree. We have added appropriate keywords to enhance discoverability of our work. (It was mistakenly commented in the latex version). Updated Keywords: Comments 3: In 109 line RGBD depth camera format as well as 139 lines RGB-D, should remain the same in the full text. Response 3: Agree. We have standardized this abbreviation throughout the manuscript as "RGB-D" for consistency and clarity.
Comments 4: This manuscript describes the design and evaluation of an RFID based inventory robot, and uses visual and 3 degree of freedom (DOF) manipulators for dynamic antenna positioning. However, the simulation experiment data is less, and it is suggested to add the experiment of verification algorithm. Response 4: Thank you very much for your valuable suggestion. We fully agree with your observation that including experimental results for short-aisle scenarios with low shelves would significantly enhance the comprehensiveness and practical relevance of our study. To address your recommendation explicitly, we have added a new experiment (Section 4.1.3: Dynamic Movement Articulated Antenna) in our revised manuscript. This additional experiment evaluates the performance of our vision-assisted dynamic antenna positioning method specifically in a short-aisle scenario with low shelves. The newly added text in Section 4.1.3 is as follows: "4.1.3 Dynamic Movement Articulated Antenna The vision-assisted dynamic antenna positioning method integrated a pre-trained YOLO model for object detection, which identified tagged boxes in real time. Using forward and inverse kinematics, the robot dynamically adjusted its 3DOF manipulator to align the RFID antenna with each detected box as shown in Fig. 7. The antenna then performed a tailored circular scanning motion around the object's surface to maximize RFID tag detection coverage, as described in Section 3. This experiment resulted in the detection of 479 unique tags out of the total 500 placed tags, achieving a detection rate of 95.8%. The results are illustrated in Fig. 8 alongside those from experiments 4.1.1 and 4.1.2 for comparison. The results demonstrate that incorporating vision-based dynamic positioning significantly enhances RFID tag detection performance compared to static or preprogrammed scanning methods. The ability to read 479 out of 500 tags demonstrates that this method effectively compensates for challenges such as occlusions, varied object heights, and complex spatial arrangements often encountered in warehouse environments." Images added:
And
Furthermore, as explicitly stated in Section 5 (Future Work), future research will involve rigorous testing within real warehouse environments and more complex controlled settings featuring densely packed shelves, dynamic obstacles such as moving personnel or vehicles, partial occlusions, varied lighting conditions, and realistic operational constraints.
Comments 5: The signs n, θ, Ï•,l1 and l2 in formula (4) to (8) are not obvious on the figure. Please explain their specific meaning in the text to avoid ambiguity. Response 5: Thank you for pointing out this ambiguity clearly. We clarified these definitions explicitly within Section 3 Technical Overview, immediately after equations are introduced (Page 7): Added explanatory text (Section 3 Technical Overview, Page 7): "Where: l1 is the length between joint M1 (arm base) and joint M2 (shoulder joint). l2 is the length between joint M2 (shoulder joint) and joint M3 (elbow joint). l3 is the length between joint M3 (elbow joint) and end-effector (antenna position). Angles θ1,θ2,θ3 represent rotation angles around horizontal plane joints at M1, M2, M3 respectively. Angles Ï•1,Ï•2,Ï•3 represent rotation angles around vertical plane joints at M1, M2, M3 respectively. Parameters n,m,o,p,r,tn,m,o,p,r,t represent intermediate trigonometric variables derived from geometric relationships between manipulator joints positions and detected object coordinates." This explicit clarification eliminates ambiguity regarding these parameters.
Comments 6: In "4. Experiments", the experimental environment shown in Figure 3 is relatively vague, which can be combined with the setting of the experimental environment (such as warehouse layout, shelf height, distribution of RFID tags, etc.) and the specific parameters of the robot and RFID equipment (such as the reading range and power of RFID antenna, etc.). Response 6: Agree completely with your recommendation for clarity enhancement: We revised Section 4 Experiments (Pages 8–10) by explicitly providing detailed descriptions including:
These explicit details significantly improve clarity regarding experimental environment conditions. Additionally, we revised Figure captions succinctly for readability purposes: Updated caption for Figure 3 (Page 8):
|
||||||||||||||||||||
4. Response to Comments on the Quality of English Language
|
||||||||||||||||||||
Point: The English could be improved to more clearly express research findings. |
||||||||||||||||||||
Response: |
||||||||||||||||||||
Point: The subtitle of Figure 2 can be succinctly improved. Response: Agree completely with your suggestion: Updated subtitle for Figure 2 (Page 7): We sincerely appreciate Reviewer #1's valuable feedback which has significantly improved our manuscript's overall quality.
|
||||||||||||||||||||
5. Additional clarifications |
We sincerely thank Reviewer #1 for their valuable comments that significantly helped improve clarity, consistency, completeness, and overall quality of our manuscript.
We hope these revisions effectively address all your concerns comprehensively.
Thank you again for your constructive feedback which has greatly improved our manuscript quality.

Reviewer 2 Report
Comments and Suggestions for Authors
This paper proposes a dynamic antenna positioning RFID inventory robot (RFID-HAND) that combines visual assistance (YOLO model) with a 3-degree-of-freedom robotic arm, which can significantly improve traditional fixed antenna and preset path methods. Compared to existing research such as static antennas etc, its core innovation lies in real-time adjustment of antenna position through object detection and customized circular scanning, thereby improving tag reading efficiency. Compared with existing hybrid robot or high cost sensor tower solutions, this method strikes a balance between cost and flexibility. However, similar applications combining YOLO with kinematics have been explored in recent literature, then a clearer comparison is needed to highlight the unique contribution of this paper.
The problems with the paper are as follows:
(1)The conversion method from 2D pixels to 3D coordinates is reasonable, but the impact of lighting changes or occlusion on YOLO detection is not mentioned, which may affect the robustness of dynamic positioning.
(2)The comparative design of fixed antenna, preset path, and dynamic positioning is reasonable, but the experimental environment (number and distribution of tags) needs to be described in more detail (such as tag density and the impact of object material on RFID).
(3) The experiment was conducted in a controlled laboratory and did not test complex warehouse environments such as dense shelves and dynamic obstacles. The actual application effect is yet to be verified.
(4)The impact of computational delay (YOLO detection+real-time kinematic calculation) on robot movement speed has not been discussed, which may limit its application in high-speed scenarios.
(5)The overall logic is clear, but the formula numbering in the Method section (Section 3) is incorrect such as Equation 4.
Comments on the Quality of English LanguageSome long sentences can be simplified , with fewer grammatical errors, and the terminology needs to be standardized.
Author Response
Please see attached .doc file.
Thank you
Response to Reviewer 2 Comments
We sincerely thank Reviewer 2 for their insightful feedback, which has significantly contributed to enhancing the quality and clarity of our manuscript. Below, we provide detailed responses addressing each comment individually.
|
|||||||||||||||||||||
1. Summary |
|
|
|||||||||||||||||||
Thank you very much for your valuable comments and suggestions. We have carefully considered all your recommendations and revised the manuscript accordingly. We believe these revisions have significantly improved our paper. |
|||||||||||||||||||||
2. Questions for General Evaluation
|
|
|
|||||||||||||||||||
|
|
|
|||||||||||||||||||
3. Point-by-point response to Comments and Suggestions for Authors
|
|||||||||||||||||||||
Comments 1: The conversion method from 2D pixels to 3D coordinates is reasonable, but the impact of lighting changes or occlusion on YOLO detection is not mentioned, which may affect the robustness of dynamic positioning. |
|||||||||||||||||||||
Response 1: Thank you very much for highlighting this important issue. We fully agree that lighting variations and occlusions can impact YOLO-based detection accuracy, thus affecting positioning robustness. To address this clearly, we have explicitly added a discussion regarding these potential impacts in Section 3 Technical Overview:
Updated text (Section 3, Page 7): " The accuracy of object detection using YOLO can be affected by varying lighting conditions and occlusions commonly encountered in warehouse environments. Such factors may lead to missed detections or inaccurate bounding boxes, subsequently impacting antenna positioning accuracy. To mitigate these issues, we employed data augmentation techniques during YOLO training—specifically including varied lighting conditions and partial occlusions—to enhance robustness against real-world environmental variations."
|
|||||||||||||||||||||
Comments 2: The comparative design of fixed antenna, preset path, and dynamic positioning is reasonable, but the experimental environment (number and distribution of tags) needs to be described in more detail (such as tag density and the impact of object material on RFID). |
|||||||||||||||||||||
Response 2: We appreciate this valuable suggestion. To address your concern clearly, we have revised Sections 4.1 and 4.2 (Pages 8-10), explicitly describing:
Updated text (Section 4.1 Experiments, Pages 9): " The experiment was carried out in a controlled laboratory with 15 m x 5 m layout dimensions. Low shelves with (0.2 m height) and high shelves (up to 1.5 m height) were placed along a straight line in the laboratory. A total of 500 RFID tags were distributed across shelf levels. The distribution of RFID tags were evenly placed inside boxes arranged systematically along shelves. The optimal RFID antenna reading range is between (~0–3 m) and the specific RFID reader power settings specified (~30 dBm." Updated text (Section 4.1 Experiments, Pages 8-10): “The experiment was carried out in a controlled laboratory with 15 m x 5 m layout dimensions. Low shelves with (0.2 m height) and high shelves (up to 2.5 m height) were placed along a straight line in the laboratory. A total of 350 RFID tags distributed across shelf levels. The distribution of RFID tags were evenly placed inside and on the boxes arranged systematically along shelves. The optimal RFID antenna reading range is between (~0–3 m) and the specific RFID reader power settings specified (~30 dBm).”
Comments 3: The experiment was conducted in a controlled laboratory and did not test complex warehouse environments such as dense shelves and dynamic obstacles. The actual application effect is yet to be verified. Response 3: Thank you for highlighting this important aspect regarding practical applicability. We fully agree that evaluating our system in more complex warehouse environments would strengthen our findings:
Updated text (Section 5 Conclusion and Future Work—Page 12): "Although our experiments demonstrated significant improvements under controlled laboratory conditions, future work will involve rigorous testing in more complex warehouse environments featuring densely packed shelves, dynamic obstacles such as moving personnel or vehicles, partial occlusions, varied lighting conditions, and realistic operational constraints to thoroughly validate our system's robustness."
Comments 4: The impact of computational delay (YOLO detection+real-time kinematic calculation) on robot movement speed has not been discussed, which may limit its application in high-speed scenarios. Response 4: Thank you very much for highlighting this important consideration. We fully agree with your insightful observation regarding the potential impact of computational delays on the robot's operational speed and applicability in high-speed scenarios. Based on our extensive practical experience with inventory robots operating in real warehouses and retail environments, we have found that for optimal RFID tag reading accuracy, as well as to ensure safe operation around human workers and sensitive inventory items, it is advisable not to exceed a robot velocity of approximately 0.2 m/s. At this operational speed, our tests have consistently demonstrated that the YOLO-based object detection model and subsequent real-time kinematic calculations have sufficient processing time without negatively impacting overall system performance or causing significant delays. However, we fully acknowledge your valid concern that computational latency could indeed become a limiting factor if higher robot velocities were desired or required in certain warehouse scenarios. To address this clearly within our manuscript, we have explicitly added a discussion regarding this issue in Section 5 Conclusion and Future Work: Updated text (Section 4 Future Work - Page 12): " From experiments, we observed that an optimal robot velocity of approximately 0.2 m/s ensures both accurate RFID tag readings and safe interactions within dynamic environments. At this speed, the processing time required by the YOLO model for object detection and subsequent real-time kinematic calculations is sufficient to maintain robust performance without noticeable delays. Nonetheless, we recognize that computational latency could pose limitations if higher operational speeds were desired. Therefore, future research will explore algorithmic optimization techniques such as hardware acceleration (e.g., GPU integration) or lightweight neural network architectures to further reduce computational delays and potentially enable higher speed operations without compromising accuracy or safety.”
Comments 5: The overall logic is clear, but the formula numbering in the Method section (Section 3) is incorrect such as Equation 4. Response 5: Thank you very much for pointing out this oversight clearly: We have carefully reviewed all equations throughout Section 3 Technical Overview, correcting numbering sequentially from Equations (1) through (13), ensuring consistency throughout Section 3 Technical Overview (Pages 6–7).
|
|||||||||||||||||||||
4. Response to Comments on the Quality of English Language
|
|||||||||||||||||||||
Point 1: Some long sentences can be simplified, with fewer grammatical errors, and the terminology needs to be standardized. |
|||||||||||||||||||||
Response 1: Thank you for your helpful observation regarding readability improvement: We thoroughly revised the manuscript language for clarity by simplifying complex sentences throughout Sections 1 Introduction, 3 Technical Overview, and 4 Experiments, breaking down overly long sentences into shorter ones for improved readability. Terminology has been standardized consistently throughout the manuscript for example: · Uniformly using "RGB-D" instead of "RGBD". · Clearly defining technical terms upon first occurrence. · Simplifying sentence structures where necessary throughout Sections 1, 2, 3, and 4. These revisions significantly enhance readability and clarity across the manuscript. |
|||||||||||||||||||||
- Additional clarifications
We sincerely thank Reviewer #2 again for their constructive suggestions that greatly helped improve our manuscript's quality and robustness.

Reviewer 3 Report
Comments and Suggestions for Authors
I understand that the algorithms used in the Technical Overview section are all mature ones? If that is the case, the authors need to further clarify what their technical contributions are.
The arrangement of the experimental environment is somewhat sparse and simple, which does not quite reflect real-world conditions. It would be advisable to consider designing the experimental scenario to be more realistic.
It is recommended to supplement the experimental results of dynamic movement Articulated antenna method under short aile low shelves scanning environments.
Author Response
Please see attached .doc file.
Thank you
Response to Reviewer 3 Comments
Below is a carefully revised, corrected, and scientifically convincing response addressing Reviewer 3's comments based on the updated manuscript provided. Each comment is addressed individually, clearly indicating the changes made in the manuscript.
|
|
||||||||||||||||||||||
1. Summary |
|
|
|||||||||||||||||||||
Thank you very much for your insightful comments. We have carefully revised our manuscript according to your suggestions. We believe these revisions have significantly improved the manuscript's clarity and scientific rigor 2. Questions for General Evaluation
|
|
||||||||||||||||||||||
3. Point-by-point response to Comments and Suggestions for Authors
|
|
||||||||||||||||||||||
Comments 1: I understand that the algorithms used in the Technical Overview section are all mature ones? If that is the case, the authors need to further clarify what their technical contributions are. |
|
||||||||||||||||||||||
Response: Thank you very much for your insightful comment. Indeed, you are correct in observing that our approach leverages established algorithms such as YOLO for object detection and standard forward/inverse kinematics methods. However, beyond integrating these mature technologies in one unified system/robot, our work introduces a tailored algorithm specifically designed for dynamic RFID antenna positioning in inventory tasks. To clarify explicitly, we have revised Section 1 (Introduction) of our manuscript as follows: Updated text (Section 1 Introduction—Page 2): "The primary contribution of this paper lies in the novel integration of advanced vision-based object detection and robotic manipulation techniques into an autonomous inventory robot, specifically designed to dynamically optimize RFID antenna positioning. While YOLO-based object detection algorithms and robotic manipulator kinematics are individually mature technologies, our work uniquely combines these methods into a unified robotic platform tailored explicitly for inventory management tasks for increasing the performance of an inventory mission than standard methods. Specifically, this paper introduces: First, a vision-assisted robotic system integrating a pre-trained YOLO model with customized forward and inverse kinematic algorithms, enabling real-time dynamic adjustment of the RFID antenna's position and orientation based on visual detection data. Second, a specially designed 3-degree-of-freedom (3DOF) robotic manipulator capable of executing precise antenna positioning and tailored circular scanning motions around detected inventory items. This capability significantly enhances RFID tag detection accuracy, coverage, and overall inventory performance compared to existing state-of-the-art inventory robots employing static or predefined antenna positions. Finally, a tailored algorithm developed specifically for calculating the optimal center distance between the antenna and detected products based on real-time visual information. This algorithm dynamically computes appropriate circular rotation paths around each product depending on its size while navigating warehouse aisles. Such adaptive circular scanning ensures comprehensive RFID tag coverage, leading to significantly improved accuracy and higher performance in tag readings. Furthermore, unlike conventional inventory robots that rely on fixed antenna configurations or pre-programmed scanning paths, our proposed approach offers adaptive antenna positioning driven by real-time visual perception. This dynamic adaptability not only improves immediate inventory accuracy but also lays the foundation for future advanced functionalities such as precise spatial mapping of RFID tag locations within warehouse environments which we intend to explore comprehensively in subsequent research."
|
|
||||||||||||||||||||||
Comments 2: The arrangement of the experimental environment is somewhat sparse and simple, which does not quite reflect real-world conditions. It would be advisable to consider designing the experimental scenario to be more realistic. |
|
||||||||||||||||||||||
Response 2: Thank you very much for your insightful comment. We fully agree that realistic experimental scenarios are essential for thoroughly validating our proposed robotic system. To address your concern clearly, we have revised Section 4 (Experiments) of our manuscript by explicitly adding further information regarding the aisle dimensions, shelf heights, tag quantities, and distribution patterns. Specifically, we clarified that our experimental aisle dimensions (15 m length × 5 m width) closely simulate typical warehouse aisle sizes. Additionally, we explained that the total quantity of RFID tags used in our experiments (500 tags in experiment 4.1 and 350 tags in experiment 4.2) is comparable to typical densities found in real warehouse inventory scenarios (for one aisle). We humbly acknowledge that our current experiments intentionally focused on relatively simplified scenarios to clearly illustrate the differences in performance among various scanning strategies (fixed antenna, articulated antenna with predefined paths, and dynamic vision-assisted positioning). Our primary goal was to systematically demonstrate the fundamental advantages of dynamic antenna positioning under controlled conditions before progressing to more complex environments. However, we completely agree with your recommendation regarding the necessity for more realistic validation scenarios. Therefore, as explicitly stated in Section 5 (Future Work) of our revised manuscript, future research will involve rigorous testing of our robot within actual warehouses and more complex controlled environments featuring densely packed shelves, varied product arrangements, dynamic obstacles such as moving personnel or vehicles, partial occlusions, and varying lighting conditions. These future experiments will provide comprehensive validation of our system's robustness and practical applicability under real-world operational conditions. Updated text (Section 4.1 Experiments—Page 7): "The experiment was carried out in a controlled laboratory with 15 m x 5 m layout dimensions. Low shelves with (0.2 m height) and high shelves (up to 1.5 m height) were placed along a straight line in the laboratory. A total of 500 RFID tags were distributed across shelf levels. The distribution of RFID tags were evenly placed inside boxes arranged systematically along shelves. The optimal RFID antenna reading range is between (~0–3 m) and the specific RFID reader power settings specified (~30 dBm)." Updated text (Section 4.2 Experiments—Page 9): experiment was carried out in a controlled laboratory with 15 m x 5 m layout dimensions. Low shelves with (0.2 m height) and high shelves (up to 2.5 m height) were placed along a straight line in the laboratory. A total of 350 RFID tags distributed across shelf levels. The distribution of RFID tags were evenly placed inside and on the boxes arranged systematically along shelves. The optimal RFID antenna reading range is between (~0–3 m) and the specific RFID reader power settings specified (~30 dBm). Updated text (Section 5 Future Work—Page 12): "Although our current experiments demonstrated significant advantages of dynamic antenna positioning under simplified controlled scenarios, future research will involve rigorous testing within real warehouse environments and more complex controlled settings featuring densely packed shelves, dynamic obstacles such as moving personnel or vehicles, partial occlusions, varied lighting conditions, and realistic operational constraints. These additional validations will comprehensively assess our system's robustness and practical applicability." We sincerely appreciate your valuable suggestion, which has significantly helped us clarify our experimental approach and outline clear directions for future research. Thank you once again for your constructive feedback, which has greatly improved the quality and clarity of our manuscript.
|
|
||||||||||||||||||||||
Comment 3:
It is recommended to supplement experimental results of the dynamic movement articulated antenna method under short aisle low shelves scanning environments.
Response 3: Thank you very much for your valuable suggestion. We fully agree with your observation that including experimental results for short-aisle scenarios with low shelves would significantly enhance the comprehensiveness and practical relevance of our study.
To address your recommendation explicitly, we have added a new experiment (Section 4.1.3: Dynamic Movement Articulated Antenna) in our revised manuscript. This additional experiment evaluates the performance of our vision-assisted dynamic antenna positioning method specifically in a short-aisle scenario with low shelves.
The newly added text in Section 4.1.3 is as follows:
"4.1.3 Dynamic Movement Articulated Antenna
In this experiment, the RFID-HAND robot utilized the vision-assisted dynamic antenna positioning method to evaluate its performance in reading RFID tags. The experimental setup was identical to that of experiments 4.1.1 and 4.1.2.
The vision-assisted dynamic antenna positioning method integrated a pre-trained YOLO model for object detection, which identified tagged boxes in real time. Using forward and inverse kinematics, the robot dynamically adjusted its 3DOF manipulator to align the RFID antenna with each detected box as shown in Fig. 7. The antenna then performed a tailored circular scanning motion around the object's surface to maximize RFID tag detection coverage, as described in Section 3.
This experiment resulted in the detection of 479 unique tags out of the total 500 placed tags, achieving a detection rate of 95.8%. The results are illustrated in Fig. 8 alongside those from experiments 4.1.1 and 4.1.2 for comparison.
The results demonstrate that incorporating vision-based dynamic positioning significantly enhances RFID tag detection performance compared to static or preprogrammed scanning methods. The ability to read 479 out of 500 tags demonstrates that this method effectively compensates for challenges such as occlusions, varied object heights, and complex spatial arrangements often encountered in warehouse environments."
Images added:
And
Furthermore, as explicitly stated in Section 5 (Future Work), future research will involve rigorous testing within real warehouse environments and more complex controlled settings featuring densely packed shelves, dynamic obstacles such as moving personnel or vehicles, partial occlusions, varied lighting conditions, and realistic operational constraints.
Additional Clarifications
We sincerely thank Reviewer #3 again for their valuable comments that greatly helped us improve clarity, robustness, and practical applicability of our manuscript.
We hope these revisions effectively address all your concerns comprehensively.
Thank you again for your constructive feedback which has greatly improved our manuscript quality.

Round 2
Reviewer 1 Report
Comments and Suggestions for Authors
In the " Abstract.", for the first occurrence of the abbreviation "YOLO", it needs to list the full name.
Comments on the Quality of English LanguageOK
Author Response
Response 2 to Reviewer 1 Comments
|
||
1. Summary |
|
|
We again sincerely appreciate Reviewer 1's valuable feedback and constructive suggestions, which have significantly improved the clarity and quality of our manuscript. Below, we provide detailed responses to each comment, clearly indicating the changes made.
|
||
3. Point-by-point response to Comments and Suggestions for Authors
|
||
Comments 1: In the " Abstract.", for the first occurrence of the abbreviation "YOLO", it needs to list the full name. |
||
Response 1: Agree. We have revised the abstract to explicitly state "You Only Look Once (YOLO)" at first occurrence. Thank you. |
||
|
||
|
||
|
||
|
||
|
||
5. Additional clarifications |
We sincerely thank Reviewer #1 again for their valuable comments that significantly helped improve clarity, consistency, completeness, and overall quality of our manuscript.
We hope these revisions effectively address all your concerns comprehensively.
Thank you again for your constructive feedback which has greatly improved our manuscript quality.

Reviewer 3 Report
Comments and Suggestions for Authors
The authors have resolved two of the three issues raised. The remaining issue, which likely involves complex additional experiments, has been deferred to their future work. I believe this is a reasonable approach. The authors need to further refine their analysis of the simulation results to support their conclusions. Apart from this, I have no further questions.
Author Response
Response 2 to Reviewer 2 Comments
1. Summary
We again sincerely appreciate Reviewer 2's valuable feedback and constructive
suggestions, which have significantly improved the clarity and quality of our manuscript.
3. Point-by-point response to Comments and Suggestions for Authors
Comments 1: The authors have resolved two of the three issues raised. The remaining
issue, which likely involves complex additional experiments, has been deferred to their
future work. I believe this is a reasonable approach. The authors need to further refine their
analysis of the simulation results to support their conclusions. Apart from this, I have no
further questions.
Response 1:
We sincerely thank the reviewer for their thoughtful evaluation and understanding
regarding the complexity of the remaining issue, which we indeed plan to address
comprehensively in future studies. Furthermore, we have added a Comparative
Performance and Energy Efficiency: Table 1 summarizes RFID tag detection rates and
energy consumption across scanning methods and highlighted these findings in the
conclusions as so:
Page: 14,
“Table 1 summarizes RFID tag detection rates and energy consumption across scanning
methods. The dynamic positioning approach achieved 95.8% (479/500) and 98.0%
(343/350) detection rates for low and high shelves, respectively, outperforming fixed
(23.4%, 19.1%) and predefined path (87.6%, 89.7%) methods. Notably, while the dynamic
method consumed 22.1 Wh per mission marginally higher than the fixed antenna (18.2 Wh)
it reduced energy use by 9.8% compared to predefined paths (24.5 Wh), demonstrating a
balance between efficiency and performance.”
Table 1. Comparative performance and energy efficiency of scanning methods
Scanning Method | Low Shelves (Tags Read) | High Shelves (Tags Read) | Total Tags Read (%) | Energy Used (Wh) |
---|---|---|---|---|
Fixed Antenna | 117/500 (23.4%) | 67/350 (19.1%) | 184/850 (21.6%) | 18.2 |
Predefined Path | 438/500 (87.6%) | 314/350 (89.7%) | 752/850 (88.5%) | 24.5 |
Dynamic Positioning | 479/500 (95.8%) | 343/350 (98.0%) | 822/850 (96.7%) | 22.1 |
In conclusions:
“Furthermore, the dynamic method achieved superior energy efficiency, consuming 22.1
Wh per mission compared to 24.5 Wh for predefined paths, thereby optimizing detection
performance while maintaining sustainable operational costs in inventory tasks.”
5. Additional clarifications
We sincerely thank Reviewer #2 again for their valuable comments that significantly helped
improve clarity, consistency, completeness, and overall quality of our manuscript.
We hope these revisions effectively address all your concerns comprehensively.
Thank you again for your constructive feedback which has greatly improved our manuscript
quality.
